# Biologic and Therapeutic Implications of Genomic Alterations in Acute Lymphoblastic Leukemia

**DOI:** 10.3390/jcm10173792

**Published:** 2021-08-25

**Authors:** Ilaria Iacobucci, Shunsuke Kimura, Charles G. Mullighan

**Affiliations:** 1Department of Pathology, St. Jude Children’s Research Hospital, 262 Danny Thomas Place, Memphis, TN 38105, USA; shunsuke.kimura@stjude.org; 2Comprehensive Cancer Center, Hematological Malignancies Program, St. Jude Children’s Research Hospital, 262 Danny Thomas Place, Memphis, TN 38105, USA

**Keywords:** B-ALL, *DUX4*, *IKZF1*, *PAX5*, Ph-like, *ZNF384*, *NUTM1*, T-ALL, *NOTCH1*, *BCL11B*, transcriptome, genome

## Abstract

Acute lymphoblastic leukemia (ALL) is the most successful paradigm of how risk-adapted therapy and detailed understanding of the genetic alterations driving leukemogenesis and therapeutic response may dramatically improve treatment outcomes, with cure rates now exceeding 90% in children. However, ALL still represents a leading cause of cancer-related death in the young, and the outcome for older adolescents and young adults with ALL remains poor. In the past decade, next generation sequencing has enabled critical advances in our understanding of leukemogenesis. These include the identification of risk-associated ALL subtypes (e.g., those with rearrangements of *MEF2D*, *DUX4*, *NUTM1*, *ZNF384* and *BCL11B*; the PAX5 P80R and IKZF1 N159Y mutations; and genomic phenocopies such as Ph-like ALL) and the genomic basis of disease evolution. These advances have been complemented by the development of novel therapeutic approaches, including those that are of mutation-specific, such as tyrosine kinase inhibitors, and those that are mutation-agnostic, including antibody and cellular immunotherapies, and protein degradation strategies such as proteolysis-targeting chimeras. Herein, we review the genetic taxonomy of ALL with a focus on clinical implications and the implementation of genomic diagnostic approaches.

## 1. Introduction

Acute lymphoblastic leukemia (ALL) is the most frequent childhood tumor and despite cure rates now exceeding 90% in children, outcomes for older children and adults remain poor with cure rates below 40% in those over the age of 40 [1,2,3], despite pediatric-inspired chemotherapy regimens [4]. This discrepancy is in part attributable to the different prevalence of genetic alterations across age. ALL may be of B- (B-ALL) or T-lymphoid (T-ALL) lineage, and comprises over thirty distinct subtypes characterized by germline and somatic genetic alterations that converge on distinct gene expression profiles [5,6,7,8,9,10,11,12]. These subtypes are defined by disease-initiating recurrent chromosomal gains and losses (hyper- and hypodiploidy, and complex intrachromosomal amplification of chromosome 21); chromosomal rearrangements that deregulate oncogenes or encode chimeric fusion oncoproteins, importantly often including cryptic rearrangements not identifiable by conventional cytogenetic approaches, such as *DUX4* and *EPOR* rearrangements; subtypes defined by single point mutations (e.g., PAX5 P80R or IKZF1 N159Y); subtypes defined by enhancer hijacking (e.g., *BCL11B*-rearrangements in T-ALL and lineage ambiguous leukemia) [5]; and subtypes that “phenocopy” established subtypes, with similar gene expression profile but different founding alterations (e.g., *BCR-ABL1*-like ALL and *ETV6-RUNX1*-like ALL) [7,13,14,15]. Secondary somatic DNA copy number alterations and sequence mutations are also important in leukemogenesis and treatment response, and their nature and prevalence vary according to the ALL subtype [6]. Multiple genes are associated with predisposition to ALL, including polymorphic variants in *ARID5B, BAK1, CDKN2A*, *CDKN2B*, *CEBPE*, *ELK3*, *ERG*, *GATA3*, *IGF2BP1*, *IKZF1*, *IKZF3*, *LHPP*, *MYC*, *PTPRJ*, *TP63* and the *BMI1-PIP4K2A* locus or rare mutations in *PAX5*, *TP53*, *IKZF1* and *ETV6* [16]. Several are associated with ALL subtype, for example, variants in *GATA3* have been associated with an increased risk of Philadelphia- like (Ph-like) ALL in patients of Hispanic ancestry [17], variants in *TP63* and *PTPRJ* with *ETV6-RUNX1* ALL [18] and in *ERG* with *TCF3-PBX1* ALL and African American ancestry [19,20]. A variant in the deubiquitinase gene *USP7* has been instead associated with risk of T-lineage ALL [19].

Accurate identification of the genetic abnormalities that drive ALL is important to risk stratify disease, and to guide the incorporation of molecular targeted therapeutic approaches to reduce the risk of relapse. This has been previously relied upon conventional karyotyping, fluorescence in situ hybridization (FISH) and targeted-molecular analyses. However, studies from this past decade have highlighted the importance of next generation sequencing (NGS) approaches to identify cryptic genetic rearrangements, structural DNA variation and gene expression signatures otherwise not identifiable that demand a revision of diagnostic approaches. This review describes the current genomic landscape of B- and T-ALL, highlighting their genetic characterization and diagnostic classification, clinical features, and therapeutic implications.

## 2. B-Cell Precursor Acute Lymphoblastic Leukemia

### 2.1. Previously Established Subtypes with Recurring Chromosomal Abnormalities

Prior the advent of NGS, classification of ALL has been relied on conventional karyotyping, FISH and targeted-molecular analyses for the identification of recurring chromosomal abnormalities including aneuploidy, chromosomal rearrangements and/or known gene fusions (Figure 1).

#### 2.1.1. Subtypes with Chromosomal Aneuploidy

Chromosomal aneuploidies [21], such as hyperdiploidy and hypodiploidy, are generally early initiating events acquired prenatally during fetal hematopoiesis and likely require secondary cooperating oncogenic insults to promote leukemia development [22].

High hyperdiploidy (modal number of 51–67 chromosomes, with nonrandom gains most commonly of chromosomes X, 4, 6, 10, 14, 17, 18, and 21) is present in 25–30% of ALL in children and is associated with young age (3–5 years) at diagnosis and favorable outcome [23,24]. Mutations of genes encoding mediators of Ras signaling (*KRAS*, *NRAS*, *FLT3*, *PTPN11*) and regulators of chromatin state (e.g., the histone 3 lysine 18 and 27 acetylase and transcriptional coregulator *CREBBP*, and the H3K36 methylase *WHSC1*) are frequent concomitant genetic events in high hyperdiploid ALL [23]. *CREBBP* mutations are enriched in the histone acetyl transferase domain and are selected during disease evolution [25]. As a potential mechanism for the generation of aneuploidy, hyperdiploid ALL blasts show a delay in early mitosis at prometaphase associated with defects in chromosome alignment, which lead to chromosome-segregation defects and nonmodal karyotypes [26]. Moreover, condensin complex activity is impaired, leading to chromosome hypocondensation, loss of centromere stiffness, and mislocalization of the chromosome passenger complex proteins Aurora B kinase (AURKB) and BIRC5 (survivin) in early mitosis [26]. Notwithstanding the favorable outcome of this subtype, condensin impairment suggests novel molecular targets (condensin-complex members, AURKB, or the spindle assembly checkpoint) for potential pharmacological intervention.

Hypodiploid ALL includes near haploid (24–31 chromosomes) and low hypodiploid (32–39 chromosomes) subtypes [27]. Near haploidy is present in ~2% of childhood ALL and is associated with Ras mutations (particularly *NF1*) and deletion/mutation of *IKZF3.* The gene expression profile and patterns of co-mutation (e.g., *CREBBP* and the Ras signaling pathway) are similar to high hyperdiploid ALL, suggesting a potential common origin of these two forms of leukemia. Low hypodiploidy instead is uncommon in children (~1%) but present in >10% of adults, and is characterized by deletion of *IKZF2*, *RB1*, *CDKN2A/CDKN2B* and near-universal mutations of *TP53* mutations, which are inherited in approximately half of cases and indicate that low hypodiploid ALL is a manifestation of Li-Fraumeni syndrome [28]. Duplication of the aneuploid genome, resulting in clones with 50 to 78 chromosomes, is common, with duplicated subclones present in the majority of cases. Predominance of the duplicated clone, known as masked hypodiploidy, may be misdiagnosed as high hyperdiploidy [29]. However, these states may usually be distinguished as the duplicated hypodiploid genome typically exhibits diploid and tetraploid chromosomes; in contrast high hyperdiploidy is characterized by a mixture of triploid and some tetraploid chromosomes (e.g., chromosomes 21, X). Moreover, the pattern of chromosomal losses in hypodiploid ALL is not random and chromosome 21 is never lost indicating a central role in leukemic cell fitness [27]. Hypodiploid ALL is associated with unfavorable outcome, although this is mitigated by minimal residual disease (MRD) risk-stratified therapy in several studies [30]. Moreover, for patients who achieve MRD-negative status after induction, allogeneic transplantation has been shown to be not successful in improving overall survival [31,32]. Although MRD-oriented protocols, older adults and elderly patients with low hypodiploidy do fairly poor with higher five-year cumulative incidence of relapse compared to high hypodiploid cases [33], making them candidates for different treatment approaches (e.g., immunotherapy and targeted therapies). Among those, preclinical studies have shown that hypodiploid ALL cells are sensitive to Phosphoinositide 3-kinase (PI3K) and BCL2 Apoptosis Regulator (BCL2) inhibitors [27,34].

#### 2.1.2. iAMP21

Intrachromosomal amplification of chromosome 21 (iAMP21) accounts for 1% of childhood ALL and is associated with older children (median age at diagnosis 9 years) and a low white cell count. Behind the formation of this chromosomal abnormality there is a characteristic mechanism of breakage–fusion–bridge cycles followed by chromothripsis and other complex structural rearrangements of chromosome 21 [35]. Two germline genomic alterations are associated with a markedly elevated risk of iAMP21. These are a germline Robertsonian translocation rob (15;21) and a germline ring chromosome 21 [36]. The presence of iAMP21 is associated with poor prognosis in most studies, although this has been improved with intensive treatment [37]. 

#### 2.1.3. Subtypes with Recurrent Chromosomal Translocations and/or Gene Fusions

The t(12;21)(p13;q22) translocation with the *ETV6-RUNX1* (*TEL-AML1*) fusion is the most common alteration in childhood B-ALL occurring in 20–25% of cases [38,39,40]. The *ETV6-RUNX1* fusion is considered to be a leukemia-initiating alteration which arises in utero, as demonstrated by the identification in umbilical cord blood [39] and by the prenatal monoclonal origin in identical twins [41]. The typically prolonged latency from birth to clinically manifest leukemia indicates that *ETV6-RUNX1* alone requires cooperating genetic events to induce leukemia, including deletion of the non-rearranged *ETV6* allele, focal deletion of *PAX5* and mutation of *WHSC1* [39,42,43,44].

The t(1;19)(q23;p13) translocation encoding *TCF3-PBX1* defines a subtype of 5–6% of pediatric B-ALL but only 1% of adult cases. This fusion is associated with a pre-B immunophenotype and expression of cytoplasmic immunoglobulin heavy chain and with higher peripheral blood white cell count at diagnosis [6,7,45]. Current intensive treatment has changed the historically high risk of *TCF3-PBX1* childhood ALL that was in part ascribed to central nervous system (CNS) involvement and relapse in favorable and intermediate risk cases [46,47]. *TCF3-PBX1* leukemic cells may be amenable to inhibition of pre-BCR signaling by dasatinib and ponatinib [48,49]. This approach may lead to compensatory upregulation of *ROR1* expression, and thus, concomitant inhibition of ROR1 could enhance the sensitivity of dasatinib [50]. *TCF3* and *TCF4* are also rearranged to *HLF*, and define a rare subtype of ALL (<1%) associated with an extremely poor prognosis [3,7]. *TCF3-PBX1* and *TCF3-HLF* ALL have distinct gene expression profiles and mutational landscapes [7,51]. *TCF3-HLF* ALL is associated with expression of stem cell and myeloid markers, alterations of *PAX5* (deletions) and the Ras signaling pathway [7,51] and sensitivity to therapies inhibiting BCL2 and the pre-B cell receptor [52,53], immunologic therapies [54], and to Aurora A kinase inhibitors [55]. 

Rearrangements of the mixed-lineage leukemia 1 (*MLL1*) gene (now renamed Lysine [K]-specific methyl transferase 2A or *KMT2A)* on chromosome 11q23 to over 80 different partner genes define a subtype of leukemia with lymphoid and myeloid features and poor prognosis [12,56]. It occurs predominantly in infants (~80%), with a second peak of onset in adulthood where the most common partner of rearrangement is *AFF1* [57]. It is typically associated with pro-B (CD10-) immunophenotype, and expression of myeloid markers. Irrespective of fusion partner or lineage phenotype this subtype shows a distinct gene expression signature with overexpression of *HOX* cluster genes and the HOX cofactor *MEIS1* [58,59]. In infant *KMT2A*-rearranged ALL, the PI3K and Ras pathways are commonly altered [7,60,61]. *KMT2A* rearrangement is associated with altered chromatin patterning including H3K79 methylation, which has stimulated development of novel therapeutic approaches including inhibition of DOT1L [62], bromodomain, Menin, and the polycomb repressive complex [57,63,64]. The lineage plasticity characteristic of *KMT2A*-rearranged ALL is important in the context of immunotherapy, as this may facilitate loss of expression of CD19 and escape from CD19 Chimeric antigen receptor T cell (CAR-T) therapy [65].

The frequency of patients with *BCR-ABL1* (Philadelphia chromosome) arising from the t(9;22)(q34;q11) translocation increases with age with 2–5% in childhood, 6% in adolescents and young adults (AYA), and more than 25% in adults [66,67]. Although historically considered a high-risk subtype, the incorporation of tyrosine kinase inhibitors (TKIs) into the standard treatment regimen for *BCR-ABL1*-positive ALL significantly improved clinical outcomes [68,69,70]. Secondary cooperative mutations are *IKZF1*, *PAX5* and *CDKN2A/B* deletions [42,69,71,72]. *IKZF1* alterations (most commonly deletions) have been associated with unfavorable outcome irrespective of TKI exposure [68,73], especially when co-occurring with (*CDKN2A* or *CDKN2B*, *PAX5*, or both: *IKZF1*^plus^) [68,69].

### 2.2. Emerging B-ALL Subtypes Defined by Genome Sequencing Studies

NGS approaches, particularly whole transcriptome sequencing (WTS), have enabled several research groups the identification of a large number of novel genetic alterations. These include cryptic rearrangements not identifiable by conventional approaches; novel subtypes that “phenocopy” established subtypes sharing similar gene expression profile but having different founding alterations; and subtypes defined by a single point mutation.

#### 2.2.1. *DUX4*, *MEF2D*, *ZNF384* and *NUTM1* Gene Fusions

Translocation of *DUX4* to the immunoglobulin heavy chain locus (*IGH*) is a cytogenetically cryptic alteration occurring in 5–10% of B-ALL and resulting in overexpression of a 3′ truncated DUX4 protein [7,13,74,75,76,77]. *DUX4* is located within the D4Z4 subtelomeric repeat element on chromosome 4q/10q and encodes a double homeobox transcription factor that activates expression of large number of genes in early developing embryos, but it is thereafter silenced in most somatic cells [78]. Aberrant *DUX4* expression is associated with facioscapulohumeral dystrophy (FSHD) [79], while *DUX4* rearrangements have been also identified in Ewing-like sarcoma [80] and rhabdomyosarcoma [81]. In B-ALL, truncated DUX4 protein binds to an intragenic region of *ERG* resulting in transcriptional deregulation, and commonly, expression of a C-terminal ERG protein fragment, and/or *ERG* deletion. This subtype has a very distinctive gene expression profile and immunophenotype (CD2 and CD371 positive), common deletions of *IKZF1* (40%) and despite this, excellent outcome [76,77,82,83]. Accurate identification of all cases of *DUX4*-rearranged ALL requires direct identification of rearrangement (e.g., by WTS), or alternatively, gene expression-based clustering or high *DUX4* expression. The detection of strong CD371 cell surface expression by flow cytometry is a promising surrogate marker for this subtype [84]. Although *ERG* deletion is common in, and largely restricted to *DUX4*-rearranged ALL, these deletions are secondary events, commonly subclonal, and not present in all cases. Thus, the use of *ERG* deletion as a surrogate for identification of *DUX4*-rearranged ALL is suboptimal and should be avoided.

*MEF2D* (myocyte enhancer factor 2D) rearrangements occur in ~4% of childhood and 10% adult B-ALL cases. This subtype shows a distinct immunophenotype with low/absent expression of CD10, and positivity for CD38 and cytoplasmic µ chain, and distinct expression profiles [7,85,86,87,88]. *MEF2D* is the 5′ partner in all described fusions, whereas B-cell CLL/lymphoma (*BCL*) 9 and heterogeneous nuclear ribonucleoprotein U-like 1 (*HNRNPUL1*) are the two most recurrent 3′ partners. The rearrangements result in enhanced MEF2D transcriptional activity, increased *HDAC9* expression and sensitivity to histone deacetylase inhibitors, such as panobinostat [85]. MEF2D has also been implicated in a core transcription factor regulatory circuit involving SREBF1 that regulates pre-BCR and lipid metabolism, that are therapeutic vulnerabilities [89]. Sensitivity to staurosporine and venetoclax has been also described [90]. *MEF2D*-rearranged ALL shows high levels of minimal residual disease and is considered to be an unfavorable subtype because of its poor event-free survival rates [82,83].

*ZNF384-*, or less commonly, *ZNF362*-rearranged acute leukemia is a biologically and clinically distinct leukemic subtype present in ~6% of childhood, 7.3% of adult, and 15% of AYA B-ALL, and in 48% of B/myeloid mixed phenotype acute leukemia (MPAL) [7,13,91,92,93]. These cases show a characteristic immunophenotype with weak CD10 and aberrant expression of the myeloid markers, CD13, and/or CD33 [92,94]. Expression of myeloperoxidase (MPO) is often the only feature distinguishing cases diagnosed as B-ALL (MPO−) or B/myeloid MPAL (MPO+). Different fusion partners, usually transcription factor (e.g., *TAF15* and *TCF3*) or chromatin modifiers (e.g., *CREBBP*, *EP300*, *SMARCA2*, and *ARID1B*) have been identified for ZNF384, with EP300 being the most common. In all rearrangements the zinc-finger domains of the C2H2-type zinc-finger transcription factors ZNF384/ZNF362 are retained [7,13,91,92,95]. The same cooperating genetic alterations and transcriptional profile is observed in *ZNF384*-rearranged B-ALL and MPAL, and both exhibit lineage plasticity during disease progression (e.g., with shift in immunophenotype from lymphoid to myeloid from diagnosis to relapse). *ZNF384* rearrangements are acquired in a subset of hematopoietic stem cells and prime leukemia cells for lineage plasticity [92]. A report of *ZNF384*-rearranged ALL in twins implicated a fetal hematopoietic progenitor as the cell of origin confirming that these rearrangements are founder alterations [96]. Prognosis varies by fusion partner: the *EP300-ZNF384* fusion is associated with favorable outcome while the *TCF3-ZNF384* fusion is frequently associated with late relapses and a poor prognosis [92,93]. However, overexpression of *FLT3*, characteristic of this subtype, makes this leukemia amenable to FLT3 inhibition [97]. 

*NUTM1* (nuclear protein in testis midline carcinoma family 1) rearrangements (<2% of childhood B-ALL and mostly infant without *KMT2A*-rearrangements) [7,13,88,98,99,100] are characterized by fusion of *NUTM1* to different partners, including transcription factors and epigenetic regulators (e.g., *ACIN1*, *AFF1*, *ATAD5*, *BRD9*, *CHD4*, *CUX1*, *IKZF1*, *RUNX1*, *SLC12A6*, and *ZNF618*), that drive aberrant *NUTM1* expression [7,13]. In all fusions, the NUT domain is retained, and this is hypothesized to lead to global changes in chromatin acetylation [101] and to sensitivity to histone deacetylase inhibitors or bromodomain inhibitors in case of fusions with BRD9. *NUTM1* rearrangements confer an excellent prognosis to current therapeutic approaches [82,83,98]. Since not all *NUTM1* fusions are detectable by karyotyping either break-apart FISH or, preferably, WTS are the best approaches for diagnosis. In addition, the finding that both RNA expression of the 3′ exons and protein expression are highly specific for this subtype may help in diagnosis. 

#### 2.2.2. Subtypes That Phenocopy Established Subtypes

##### Ph-Like ALL

Ph-like or *BCR-ABL1*-like ALL is characterized by a gene expression signature similar to that of Ph-positive ALL but lacking the pathognomonic BCR-ABL1 oncoprotein of Ph+ ALL [15,66,102,103,104,105,106,107,108,109,110]. Its incidence ranges from ~10–15% in children to ~20% in older adults, with a peak (25–30%) in the AYA ALL population. Similar to patients with Ph+ ALL, patients with Ph-like ALL often exhibit adverse clinical features and poor outcome and frequently harbor alterations of *IKZF1* or other B-lymphoid transcription factor genes. Over 60 heterogenous genetic alterations in kinases and cytokine receptors drive constitutively active kinase or cytokine receptor signaling, many of which have been shown to be druggable with a variety of kinase inhibitors. The most commonly mutated pathways are the ABL and JAK-STAT pathways with multiple rearrangements and lesions that converge on downstream ABL/JAK-STAT signaling. Founder alterations may be grouped into three broad types: (i) JAK/STAT alterations including: mutations activating cytokine receptors (e.g., *CRLF2* and *IL7R*); enhancer hijacking gene rearrangements deregulating cytokine receptor expression (e.g., *IGH-CRLF2* and *P2RY8–CRLF2*) [111,112,113,114]; gene fusions and/or mutations activating kinases (e.g., *JAK1*, *JAK2*, *JAK3*, *TYK2*); and rearrangements hijacking and truncating cytokine receptor expression (e.g., cryptic *EPOR* rearrangements) [115]; (ii) fusions involving ABL-class genes (*ABL1*, *ABL2*, *CSF1R*, *LYN*, *PDGFRA*, *PDGFRB*); and (iii) less common fusions (*FLT3*, *FGFR1*, *NTRK3*, *PTK2B*) [109] whose number is growing with increasing sequencing studies of different cohorts. Among these, alterations of *CRLF2* are present in approximately half of Ph-like ALL in AYAs and adults. *CRLF2* is located in the pseudoautosomal region of the sex chromosomes (*PAR1*) at Xp22.3/Yp11.3, and its alterations include: (1) a cryptic rearrangement that juxtaposes *CRLF2* to the IGH locus; (2) a focal deletion in the pseudoautosomal region of the sex chromosomes resulting in P2Y receptor family member 8 (*P2RY8*)-*CRLF2* fusion that positions *CRLF2* under the control of the *P2RY8* promoter; and less frequently by (3) an activating *CRLF2* point mutation, F232C. PAR1 deletions, as a surrogate for rearrangement of *CRLF2*, have been incorporated into the criteria for “*IKZF1*^plus^”, a designation based on DNA copy number profiling, commonly by single nucleotide polymorphism (SNP) or multiplex ligation-dependent probe amplification (MLPA) arrays. In some studies, *IKZF1*^plus^ has been associated with a higher risk of relapse defined by co-occurrence of the *IKZF1* deletion with deletion of *CDKN2A*, *CDKN2B*, *PAX5*, and/or PAR1 region in the absence of *ERG* deletion [116]. Notably, however, the *IKZF1*^plus^ designation typically does not consider cases with *IGH-CRLF2* due to the inability of these platforms to detect this alteration. 

The heterogeneous genomic landscape and often cytogenetically cryptic alterations identified in Ph-like ALL may make diagnosis of this entity and its driver alterations challenging, but several tractable diagnostic approaches are available, depending on technical capability of a laboratory, and the desired clinical/diagnostic endpoint (e.g., identification of the gene expression profile of Ph-like ALL c.f. identification of the most common driver kinase alterations). Comprehensive clinical NGS, including WTS, is the best approach to identify Ph-like ALL patients with targetable kinase alterations, as it enables analysis of gene expression, fusions, aneuploidy and sequence mutations. Selective/capture-based sequencing approaches (e.g., Archer FusionPlex, and FoundationOne Heme) also identify the majority of kinase-deregulating rearrangements in Ph-like ALL. If genomic approaches are not available, a more targeted screening approach using routine diagnostics, including flow cytometry (especially for *CRLF2*, for which positivity on flow cytometry is strongly correlated with rearrangement) and FISH for the most common kinase targets of rearrangement, is still effective for swift identification of Ph-like ALL [104]. The identification of specific genetic lesions is important for guiding targeted therapeutic intervention as a proportion of kinase-activating alterations in Ph-like ALL can, at least based on in vitro and preclinical models, be targeted by FDA-approved TKIs: JAK-STAT signaling (JAK inhibition); ABL-class fusions (ABL inhibitor); FLT3 and NTRK3 fusions (FLT3 and NTRK3 inhibitor) [104]. Several recent studies have described the efficacy of ABL1 and NTRK inhibitors in the treatment of Ph-like ALL cases with rearrangement of these genes [117,118]. Combinatorial use of kinase inhibitors against multiple signaling has shown synergism in patient-derived xenograft (PDX) models of CRLF2/JAK mutant (JAK and PI3K/mTOR inhibitors), ABL/PDGFR mutant (dasatinib and PI3K/mTOR inhibitor) and EPOR-rearranged (ponatinib and ruxolitinib) [119]. Moreover, recently dual JAK/GSPT1-degrading proteolysis-targeting chimeras PROTACs have been developed and showed efficacy in Ph-like B-ALL kinase-driven PDX models which were otherwise unresponsive to type I JAK inhibitors [120]. Lastly, the use of immunotherapeutic agents, such as blinatumomab, inotuzumab, and CAR-T cells (including those targeting CRLF2 [121], represents a promising alternative approach for this subtype which is irrespective of a specific genetic alteration or response to prior chemotherapies [104]). 

##### *ETV6*-*RUNX1*-Like ALL

*ETV6*-*RUNX1*-like ALL is characterized by a gene expression profile and immunophenotype (CD27 positive, CD44 low to negative) similar to *ETV6*-*RUNX1* ALL, but lacking the *ETV6*-*RUNX1* fusion [122] and favorable prognosis [7,13,75,122,123]. These cases harbor alternate gene fusions or copy number alterations in ETS family transcription factors (*ETV6*, *ERG*, *FLI1*), *IKZF1* or *TCF3. ETV6-RUNX1*-like ALL develops in children harboring germline *ETV6* mutations with subsequent somatic alterations of the second *ETV6* allele, consistent with the notion that biallelic alteration of *ETV6* is central in leukemogenesis [124]. It is more common children (~3%) and confers an unfavorable prognosis in children due to high levels of MRD and worst event-free survival rates [82].

#### 2.2.3. Subtypes Defined by a Single Point Mutation

##### PAX5 P80R and PAX5alt

The PAX5 P80R subtype (~3% of B-ALL cases) is characterized by the presence of a hot spot mutation at amino acid 80 in the DNA binding domain of the paired box DNA-binding transcription factor PAX5 [7,13,125,126]. B-ALL cases with PAX5 P80R show a distinct gene expression profile with the majority of cases having either hemizygous or homozygous mutation, caused by deletion of the wild-type *PAX5* allele or copy-neutral loss of heterozygosity. In a subset of cases, in addition to PAX5 P80R there is a second frameshift, nonsense or deleterious missense *PAX5* mutation. Thus, biallelic *PAX5* alterations—with mutation of one allele and loss of activity of the second allele—are a hallmark of this subtype [7,125]. In support of the role of biallelic alteration of *PAX5* in the pathogenesis of this subtype, knock-in mouse models of germline *Pax5* mutations have shown that heterozygous Pax5^P80R/+^ knock-in mice develop transplantable B-ALL, with genetic inactivation of the wildtype *Pax5* allele [7]. In contrast, Pax5^G183S/+^ knock in mice (modeling the germline PAX5 G183S mutation observed in familial ALL) show a low penetrance of ALL [127], supporting its role as a haploinsufficient tumor suppressor. Additional important cooperating lesions include *CDKN2A* loss and signaling pathway mutations, most commonly in Ras signaling genes or in the JAK/STAT pathway [7,125].

PAX5alt comprises about 7% cases with diverse *PAX5* alterations, including rearrangements, sequence mutations and focal intragenic amplifications [7]. Over 20 different partner genes have been identified with *PAX5-ETV6* being the most common. Children in this subtype are more commonly classified as high risk rather than standard risk (according to National Cancer Institute (NCI) criteria). In addition to *PAX5* alterations, recurrent genetic lesions observed in these cases include copy number losses affecting cell-cycle regulation genes such as *CDKN2A*, *RB1* and *BTG1*, B-cell development genes, transcriptional regulators and/or epigenetic modifiers (for example, *KDM6A*, *KMT2A* and *ATRX*) [7]. Both PAX5 P80R and PAX5alt subtypes are associated with intermediate to favorable prognosis [7,82,83,126].

##### IKZF1 N159Y

*IKZF1* encodes the transcription factor IKAROS, which is a member of the family of zinc finger DNA-binding proteins required for lymphoid lineage ontogeny and homeostasis [128,129]. The most common type of *IKZF1* alteration is a focal deletion occurring in 15% of ALL cases and in >50% of high risk ALL [42,72,103]. Deletions result in loss-of-function or in the dominant negative IK6 isoform and are associated with poor outcome [73,103]. In addition to deletions, missense, frameshift and nonsense mutations have been also described in pediatric high-risk B-ALL patients. Among those the missense p.Asn159Tyr mutation defines a subtype (<1% of B-ALL) with a distinct gene expression profile characterized by upregulation of genes with roles in oncogenesis (the IKZF1-interacting gene *YAP1*), chromatin remodeling (*SALL1*), and signaling (*ARHGEF28*) that are not deregulated in other subgroups of *IKZF1*-altered ALL [7,74]. In contrast to PAX5 P80R ALL, the nonmutated wild-type allele of the mutated transcription factor (here *IKZF1*) is retained [72]. As for most other missense mutations observed in IKZF1 zinc fingers, IKZF1 N159Y induces misregulation of IKZF1 transcriptional activation, in part through distinctive nuclear mislocalization and enhanced intercellular adhesion [130]. 

#### ZEB2 H1038R and IGH-CEBPE 

In unsupervised clustering of leukemic cell gene expression, cases with the H1038R mutation in *ZEB2* phenocopy the translocation t(14;14)(q11;q32) [13], which results in IGH-CEBPE fusion, suggesting a common activated pathway of leukemogenesis and defining a rare B-ALL subtype (<1%). This is associated with *NRAS* sequence mutations (>50% of cases), upregulation of *LMO1* and downregulation of *SMAD1* and *BMP2* [10]. However, neither the *IGH* or *ZEB2* mutations are unique to this group, nor do they explain all cases in this distinct gene expression and experimental validation is required to demonstrate their role as leukemogenic drivers. B-ALL with *ZEB2* mutation is associated with poor event-free survival and high relapse [131]. 

### 2.3. Prognostic Implications

The frequent and wide use of genomics to profile the landscape of ALL has allowed a tailored refinement of risk in association with standard criteria, such as MRD levels [82] (Figure 1). In childhood B-ALL, *ETV6-RUNX1*, high-hyperdiploid, and *DUX4*-rearranged B-ALL are categorized as favorable due the highest overall survival rates and the lowest relapse rates, despite elevated early MRD in *DUX4*-rearranged cases. *BCR-ABL1*, *BCR-ABL1*-like, *ETV6-RUNX1*-like, *KMT2A*-rearranged, and *MEF2D*-rearranged ALL show high levels of MRD and the worst event-free survival rates and thus are categorized to be unfavorable subtypes. The remaining subtypes including *TCF3-PBX1*, *PAX5*alt, iAMP21, hypodiploid, *ZNF384*-rearranged, *NUTM1*-rearranged, and *PAX5* P80R ALL have intermediate risk [81]. These prognostic groups have been mostly confirmed in a historic, non-MRD risk adapted trial (UKALLXII/ECOG-ACRIN E2993, NCT00002514) in adolescents and adult B-ALL cases [83] according to the following risk assignment: standard risk genotypes: *DUX4*-rearranged, *ETV6-RUNX1*/-like, *TCF3-PBX1*, PAX5 P80R, high-hyperdiploid; high-risk genotypes: Ph-like, *KMT2A-AFF1*, low-hypodiploid/near-haploid, BCL2/MYC-rearranged; and intermediate-risk genotypes: *PAX5*alt, *ZNF384*/-like, and *MEF2D*-rearranged. 

## 3. T-Cell Acute Lymphoblastic Leukemia (T-ALL)

### 3.1. Genomic Overview of T-ALL

T-ALL leukemic cells express a subset of T-cell makers (CD3, cyCD3, CD2, CD5, CD7, CD8) and arises from immature T-cell progenitors [132,133]. Pediatric T-ALL accounts for 10–15% of newly diagnosed pediatric ALL and is characterized by higher incidence in boys, high initial white blood cell counts, mediastinal mass, CNS infiltration, and slightly worse prognosis compared to B-ALL [134]. The majority of T-ALL cases may be subclassified into subtypes according to the aberrant expression and dysregulated pathways of transcription factors and oncogenes induced by leukemia-initiating alterations involving basic helix–loop–helix (bHLH) factors (*TAL1*, *TAL2*, *LYL1*), homeobox genes (*TLX1* (*HOX11*), *TLX3* (*HOX11L2*), *NKX2-1*, *NKX2-5*, *HOXA*), *LMO1*, *LMO2*, *MYB*, *BCL11B* and *SPI1* (Figure 2) [5,135,136]. These subtypes are defined with expression profiles by WTS or microarray, however, almost half of these leukemia-initiating alterations in T-ALL show intergenic breakpoints that can be missed by WTS but rescued by whole genome sequencing (WGS) [135,136]. Epigenomic analyses have also identified novel leukemia-initiating alterations in non-coding regions [5,137,138].

*NOTCH1* activating mutations and deletion of *CDKN2A/CDKN2B* loci (9p21) are found in over 70% of T-ALL cases and considered as secondary but core events in leukemogenesis [135,136,139]. Concurrent somatic mutations and copy number alterations are frequently observed in T-ALL leading to dysregulation of several cellular pathways, including JAK-STAT signaling (*IL7R*, *JAK1*, *JAK3*, *DNM2*), Ras signaling (*NRAS*, *KRAS*, and *NF1*), PI3K-AKT signaling (*PTEN*, *AKT1*, *PIK3CA PIK3CD*), epigenetic regulation (*PHF6*, *SUZ12*, *EZH2*, *KDM6A*), transcription factors and regulators (*ETV6*, *GATA3*, *RUNX1*, *LEF1*, *WT1*, *BCL11B*), and translation regulators (*CNOT3*, *RPL5*, *RPL10*) [135,136,140,141]. Accumulation of these aberrant expression and dysregulated pathways disrupt the normal T-cell differentiation, proliferation, and survival, and results in T-ALL with unique gene expression signatures reflecting the point of differentiation arrest during T-cell development [133,140]. In addition to expression profiles, DNA methylation signatures are also associated with immunophenotypic profiles and normal T-cell development differentiation stage [142,143].

### 3.2. T-ALL in Early Stages of Cortical Thymocyte Maturation

T-ALL with CD1a^+^, CD4^+^, and CD8^+^ immunophenotype includes several subgroups, such as rearrangements of *TLX1*, *TLX3*, *NKX2-1*, reflecting a differentiation arrest in early stages of cortical thymocyte maturation and confers a relatively favorable prognosis [144,145]. These subgroups almost commonly harbor *NOTCH1* and *CDKN2A* alterations. Dysregulated expression of HOX transcription factor genes is mostly induced by chromosomal translocations and inversions that juxtapose these genes to enhancers in the *TCR* and *BCL11B* regulatory regions [135,146]. Importantly, *BCL11B* rearrangements (*BCL11B*-*TLX3*) in this subgroup are mechanistically distinct from those identified in *BCL11B*-rearranged lineage ambiguous leukemias, in that in the *BCL11B-TLX3* leukemia, the *BCL11B* enhancer is used for aberrant expression of *TLX3* at the cost of the loss of expression of *BCL11B*, leading to complete difference in expression profiles [5,146,147]. Instead, *TLX3* rearranged T-ALL (including *BCL11B*-*TLX3*) shares gene expression signatures, DNA methylation profiles, somatic mutations (*BCL11B*, *WT1*, *PHF6*, *DNM2*), and downstream targets (JAK-STAT, epigenetic regulators) with *TLX1* rearranged T-ALL [135,136,142,143,145]. Some of overlapping genomic features with *TLX1*/*TLX3* rearranged T-ALL, including *NUP214-ABL1* (TKIs) and JAK-STAT pathway (ruxolitinib, a JAK-STAT inhibitor), can be targetable and have been incorporated into ongoing clinical trials [148].

### 3.3. TAL1-Driven T-ALL with Late Stages of Cortical Thymocyte Maturation

Deregulation of the *TAL1* oncogene is a feature of T-ALL that typically exhibits a late cortical thymocyte immunophenotype (CD4^+^, CD8^+^, CD3^+^) and comprises approximately 40% of T-ALL [135,136]. This T-ALL subtype includes *TAL1* and *TAL2* rearranged cases and is further classified into two subgroups by expression profiles whose one expresses *PTCRA* (pre-TCR) suggesting LCK activation that correlated with dasatinib sensitivity [136,145,149]. During normal T-cell differentiation, *TAL1* expression is transcriptionally silenced along with T-cell lineage commitment to proceed appropriate *TCR* rearrangements and differentiation [133]. *TAL1* overexpression is induced by several mechanisms: (1) chromosomal translocations with *TCRA/D*; (2) sub-microscopic interstitial deletion (*STIL-TAL1*); (3) disruption of insulated neighborhoods by losing CTCF binding sites [150]; and (4) somatic indels in a noncoding intergenic regulatory element upstream of *TAL1* to generate aberrant MYB binding site (*MuTE*) [137]. The latter two mechanisms have benefited of NGS technologies for their identification. Dysregulated *TAL1* expression inhibits the function of E-protein dimers by forming TAL1-E-protein heterodimer [151]. Furthermore, TAL1 forms the central node of the core regulatory circuit to coordinately regulate downstream target genes with several hematopoietic transcription factors including *GATA3*, *RUNX1*, *MYB*, and the ETS family genes, which is active in normal hematopoietic stem cell (HSC) and progenitor cells [152,153], and RUNX1 inhibition is reported to impair the growth of T-ALL but not normal hematopoietic cells [154]. However, although TAL1 functions as a master transcription factor related to T-cell differentiation and leukemogenesis of T-ALL, only 30% of transgenic mice develop T-ALL after a latent period, indicating that additional abnormalities are required for leukemogenesis [155]. Expression of *Lmo2* accelerates the onset of leukemia in *Tal1* transgenic mice, and *LMO1/LMO2* are commonly expressed in human *TAL1*-driven T-ALL [156,157]. Other cooperative genes and noncoding RNAs in *TAL1*-driven T-ALL include *ARID5B*, *ARIEL*, and *MYC*, driving aberrant expression of *TAL1* [158,159]. In addition, PI3K-AKT pathway genes including *PTEN* are frequently mutated in this subgroup [135,136], which associates with glucocorticoid resistance and can be reversed by the inhibition of this pathway [160]. Several cell cycle regulators including *CDK6* and *CCND3* are regulated by TAL1 complex [152] and may be potential targets of therapeutic intervention [161].

### 3.4. Early T-Cell Precursor (ETP) ALL and Mixed Phenotype Acute Leukemia

ETP-ALL is often referred to as a subtype of T-ALL as it exhibits an immunophenotype analogous to the earliest stages of T-cell development (cytoplasmic CD3^+^, CD7^+^; CD8^−^, CD1a^−^, CD5^weak^), and with expression of myeloid and/or stem-cell markers [144,162]. However, the genomic alterations and gene expression profile of ETP-ALL are more similar to a hematopoietic stem cell than a T cell precursor, suggesting that ETP-ALL could be included in a subgroup of immature acute leukemias of ambiguous lineage (ALAL), originating from a hematopoietic progenitor at a maturational stage prior to initiation of a definitive program of T cell differentiation. Consistent with this, recent studies have defined a subgroup of *BCL11B*-deregulated ALAL, that includes one third of ETP-ALL and T/myeloid mixed phenotype acute leukemia (T/M MPAL) cases with a very distinct expression profile [5]. *BCL11B*-deregulated ALAL is characterized by structural variations of the region containing *BCL11B* at 14q32 including translocations and high-copy amplification generating a distal neo-enhancer, that each leads to aberrant expression of *BCL11B*, in the case of the rearrangements by hijacking super-enhancers active in CD34+ hematopoietic stem and progenitor cell (HSPCs) [5,147]. *FLT3* activating mutations were found in 80% of *BCL11B*-deregulated ALAL, and concurrent expression of *BCL11B* and FLT3-ITD on HSPC exhibited synergistic effects on activating T-cell directed differentiation to express cytoplasmic CD3 while blocking myeloid differentiation [5]. Other genomic features of ETP-ALL include a subgroup of aberrant expression of PU.1 (*SPI1* fusions), *HOXA* genes (rearrangements of *HOXA* genes, *KMT2A* rearrangements, *PICALM*-*MLLT10*, *SET*-*NUP214*) and mutations of multiple cellular pathways (Ras signaling, JAK-STAT signaling, and epigenetic regulators) and transcription factors related to T-cell development [135,136,163]. Especially, T-ALL with *SPI1* fusions represents unique expression profiles with high relapse rate [5,136]. Again, several of these genomic mutations were shared with T/M MPAL, including biallelic *WT1* alterations, mutations of hematopoietic transcription factors (*ETV6*, *RUNX1*, *CEBPA*) and activating mutations of signaling pathways (JAK-STAT, *FLT3*, Ras) [92,163], supporting that they are similar entities in the spectrum of immature leukemias and both might have sensitivity to FLT3 and/or JAK inhibition [164].

### 3.5. NOTCH1 Activating Mutations in T-ALL

*NOTCH1* encodes a highly conserved ligand-dependent transcription factor. The NOTCH1 signaling pathway plays an important role in the commitment of T-cell lineage specification and for further T-cell development [133,165]. In T-ALL, NOTCH1 activating mutations are found in more than 70% of cases and is considered an oncogene involved in leukemogenesis [135,136]. Aberrant activation of NOTCH1 pathway in T-ALL is mostly induced by (1) ligand-independent activation (somatic mutations, indels and large deletions that disrupt the negative regulatory region), or (2) impairment of the proteasomal degradation of intracellular domain of NOTCH1 (truncation of the PEST domain, *NOTCH1* mutations in 3′ untranslated region, and *FBXW7* mutations) [166,167,168,169,170,171]. These two types of *NOTCH1* activating mutations have synergistic effects and more than 20% of T-ALL cases harbor both types of alterations [166]. However, most *NOTCH1* activating mutations found in human T-ALL are considered as a weak tumor initiator event. Co-existence of both types of *NOTCH1* mutations in hematopoietic progenitors tends to induce a transient preleukemic CD4^+^/CD8^+^ double positive cells and takes 10 to 15 weeks to fully transform into T-ALL, suggesting that they are alone incompletely leukemogenic [172,173,174]. In addition, more than 40% of T-ALL cases harbor subclonal *NOTCH1* activating mutations and their heterogeneity at diagnosis was reported by several studies [135,136,175]. Furthermore, *NOTCH1* activating mutations are considered to be acquired as a late secondary event in leukemogenesis [139,175,176].

A key target of NOTCH1 is the *MYC* oncogene that shares several overlapping target genes with NOTCH1 to promote cell proliferation and dysregulate anabolic pathways in T-ALL [174,177,178]. NOTCH1 controls T-cell-specific distal enhancer of *MYC* (“NMe”), resulting in the NOTCH1-MYC regulatory circuit [174,177,178]. In addition, pre-TCR signaling also correlates with NOTCH signaling, leading to LCK signaling and robust cell growth at DN3 stage in the T-cell development, which can be targetable by dasatinib [149,179]. 

Due to the high prevalence and importance of *NOTCH1* activating mutations in T-ALL, targeted therapy on NOTCH1 pathway has been a major interest. This includes γ-secretase inhibitors (GSIs), ADAM inhibitors, SERCA inhibitors, and monoclonal antibodies [180]. Among them, GSIs, that block the activation process of NOTCH receptors by inhibiting proteolytic cleavage, have been tested in preclinical and Phase 1 studies [181,182]. However, the usage of GSIs in T-ALL is still in a developing phase due to gastrointestinal toxicity and insufficient antitumor responses that mostly induce transient growth arrest rather than cell death [183,184]. To overcome these problems, combination with other agents have been explored including glucocorticoids that showed synergistic effects by reversing glucocorticoid resistance [185]. Inhibition of mTORC1 signaling and PKCδ signaling are also promising combination strategies to restore GSIs sensitivity in resistant cells [186,187].

## 4. Implications for Diagnosis

The revolution in genomic characterization of ALL has created important opportunities and challenges for the clinical implementation of sequencing-based approaches for diagnosis and management of ALL (Table 1). This is particularly true for B-ALL, where many of the recently identified subtypes are associated with prognosis (even in the context of MRD-based risk-adapted therapy) [82,83] and where molecular characterization is needed to identify patients suitable for targeted therapy (an exemplar being Ph-like ALL). This is currently less compelling for T-ALL where identification of founding lesions driving T-ALL subtypes are of biological and mechanistic interest but are not typically used to risk stratify or guide therapy, exceptions possibly being kinase inhibition for JAK-STAT alterations and *ABL1* rearrangements, identification of alterations in Ras, *PTEN*, *NOTCH1* and/or *FBXW7* that have been found to be associated with outcome in some studies [188], and LCK dependence for dasatinib therapy [149]. The challenge is clinical implementation of appropriately comprehensive diagnostic approaches to identify all key genomic features. Despite the mutationally sparse genome of ALL, there is striking diversity of the nature of underlying driver alterations, including sequence mutations, DNA copy number alterations, and structural variations, many of which may involve the non-coding genome. Accurate subtyping is also challenged by the inability of conventional cytogenetic and targeted molecular approaches to identify several types of driver (e.g., *DUX4*-rearrangement) and the importance of identifying phenocopies (e.g., *ETV6-RUNX1*-like, and Ph-like ALL). Thus, moving forward, optimal clinical diagnostics require genomic approaches. The choice of approach in part rests on how clinical information will be used. If comprehensive subtyping and identification of all potentially clinically relevant genomic alterations is desirable, a combination of DNA and RNA-based technologies is required. For example, the combination of WGS and WTS enables the identification of sequence mutations, DNA copy number alterations, aneuploidy and structural variants (from WGS) together with identification of fusion chimeras, mutant allele expression, and gene expression profiling (from WTS). The use of one or both approaches is becoming increasingly widely used, and at St Jude Children’s Research Hospital, three platform sequencing (WGS, WTS and exome sequencing) is clinical standard of care, informs clinical decision making in ALL [148], and retrieves more actionable clinical information than any single platform alone [189]. WGS is offered using a paired non-tumor sample to aid identification of somatic variants and provides the opportunity to return clinically relevant germline findings. Moreover, this comprehensive approach enables a more streamlined workflow [190,191,192], provided the demands of analysis and interpretation can be met.

However, WGS is not yet widely used clinically, and many clinicians and providers seek alternative approaches to identify clinically relevant alterations. These fall into three main categories: single platform sequencing, sub-genomic sequencing, and targeted detection of genomic alterations. In the first category, single platform WTS provides near comprehensive characterization of clinically relevant alterations in ALL, particularly B-ALL: gene expression-based profiling to identify subgroups and phenocopies; fusion transcripts; and interrogation of specific sequence mutations (e.g., JAKs, PAX5 and IKZF1) [7,193]. Moreover, several methods are available that utilize expression and mutant allele fraction to robustly identify large scale chromosomal copy number changes, thus providing a surrogate for conventional cytogenetic identification of aneuploidy [7,194]. WTS as a single platform has limitations—it is challenging to identify all sequence variations although analytic platforms are improving, it cannot identify focal DNA copy number alterations that may impact targetable pathways (e.g., *SH2B3* deletions in JAK-STAT-driven Ph-like ALL) and does not identify rearrangements that may deregulate oncogenes without resulting in a RNA chimera—for example rearrangements of oncogenes in T-ALL such as *TLX3* and those involving TCR, where breakpoints are frequently intergenic [135], the diverse rearrangements in *BCL11B*-rearranged ALAL [5], or non-coding sequence mutations that drive oncogenes such as *TAL1* and *LMO1/2* [137,138]. 

Several platforms are available for targeted DNA and/or RNA sequencing, often using capture-based approaches. These including Foundation Medicine [195] and the FusionPlex ALL Kit (Invitae, previously ArcherDx). These have the advantage of being somewhat simpler to access or implement in routine diagnostic laboratories, and the ability to detect the majority of chimeric fusion events in B-ALL. Similar reservations to WTS apply regarding the limited ability of these platforms to detect intergenic rearrangements in ALL; moreover, these platforms either have limited (Foundation) or no (Archer) capability to detect DNA copy number alterations, particularly those that are single copy, and may have difficulty resolving complex rearrangements (e.g., truncating rearrangements of *EPOR* in Ph-like ALL) [115]. Capture based DNA sequencing for sequence mutations is widely used in hematological malignancies, but is not well suited to diagnosis of ALL due to the lack of detection of rearrangements and structural variations. As described above, the MLPA platform is widely used by several groups to identify focal DNA copy number alterations and the “*IKZF1*^plus^” composite genotype, but this platform is not an adequate surrogate for sensitive detection of several key subtypes: e.g., *ERG* deletion in *DUX4*-rearranged ALL (only ~50% of cases have clonal *ERG* deletion), and PAR1 deletion in *CRLF2*-rearranged ALL (*IGH-CRLF2* is usually not accompanied by PAR1 deletion). 

In the absence of sequencing-based approaches, several subtypes and drivers may be identified by flow cytometry, immunophenotypic and targeted molecular approaches. Flow cytometry may be used to detect CRLF2 rearrangements, that result in cell surface expression of *CRLF2*, as well as markers associated with distinct subtypes (e.g., CD371 in *DUX4*-rearranged ALL). FISH may be used to detect rearrangement of the most commonly rearranged genes in Ph-like ALL for which targeted therapies are currently available (e.g., *ABL*-family kinase genes, *CRLF2*, *NTRK3*) with caveats—for example, the focal insertions of *EPOR* into *IGH* and similar enhancer regions are not robustly detected by FISH due to the small size of the *EPOR* insertion. Specific subtype-defining rearrangements may be detected by conventional molecular approaches such as RT-PCR. Thus, these composite approaches may be suitable to detect many actionable alterations in ALL, but do not provide a pathway to comprehensive identification of all driver lesions of prognostic significance. 

## 5. Conclusions

Large-scale integrative genome-wide sequencing studies have profoundly transformed the molecular taxonomy of ALL, resulting in the identification of new entities with prognostic and therapeutic significance. There are over 30 different B/T-ALL subtypes defined by distinct constellations of somatic and/or germline genetic alterations that converge on distinct gene expression patterns. The identification of these dysregulated pathways is crucial for clinical management of ALL patients and most importantly for guiding therapeutic intervention. The best example is provided by the constitutively active kinases in Ph-like which are druggable by a variety of single or combinatorial TKIs. Although the enormous clinical and genetic progress of the past decade, much work remains, as most studies have lacked NGS and have not validated the mechanisms by which fusions/mutations cooperate in leukemogenesis, and not fully defined potential for targeting. Due to the heterogeneity of genetic lesions, optimal clinical diagnosis of ALL requires genomic and/or transcriptomic sequencing in order to identify fusions, aneuploidy and sequence mutations required for disease stratification. The use of such approaches is becoming increasingly widespread. Recently, new immunotherapeutic agents (e.g., developed antibodies and CAR-T cells) have been efficacious in a proportion of patients, but failed in others. Thus, efforts should be focused in the future on defining subtype specific vulnerabilities to improve treatment strategy and outcome.

## Figures and Tables

**Figure 1 jcm-10-03792-f001:**
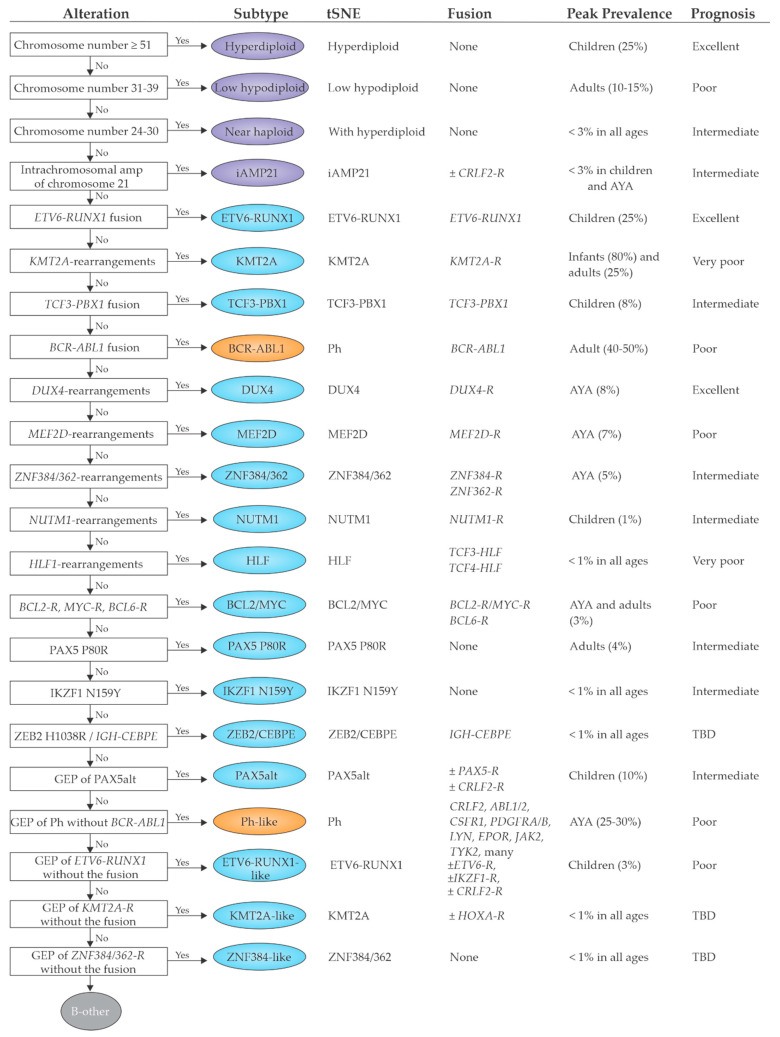
This schematic algorithm for B-ALL subtyping was modified from the figure originally published in Paietta E. et al. Molecular Classification Improves Risk Assessment in Adult BCR-ABL1-negative B-ALL. Blood Prepublished Apr 25 2021; doi:10.1182/blood.2020010144 [83]. This figure describes each B-ALL subtype according to the specific genetic alterations and gene expression profile. Moreover, for each subtype peak prevalence and prognosis are shown. Subtypes are colored according to defining genetic alteration: gross chromosomal abnormalities (purple), transcription factor rearrangements (blue), other transcription factor alterations (blue), and kinase alterations (orange). Abbreviations: AYA, adolescent and young adult; tSNE, t-distributed stochastic neighbor embedding; TBD: to be defined; -R: rearranged.

**Figure 2 jcm-10-03792-f002:**
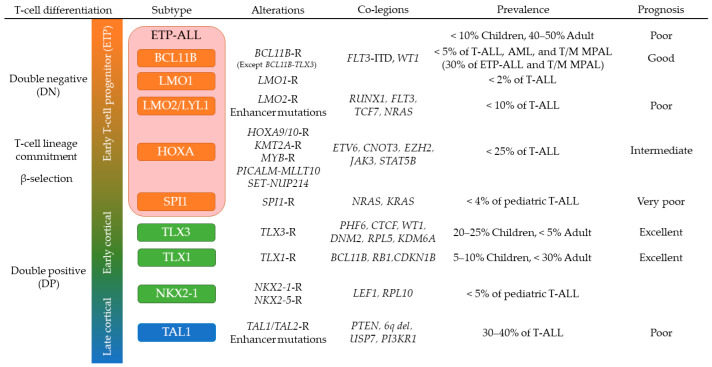
T-cell differentiation and T-ALL subtyping. This schema describes differentiation stages of each T-ALL subtype according to the specific genetic alterations leading to aberrant expression of rearranged or mutated genes. Prevalence and prognosis of each subtype are shown. Subtypes are colored according to corresponding normal T-cell differentiation stage: early T-cell precursor (ETP, red and orange), early stages of cortical thymocytes maturation (green), and late stages of cortical thymocytes maturation (blue). Abbreviations: T-ALL: T-cell acute lymphoblastic leukemia; T/M MPAL: T/myeloid mixed phenotype acute leukemia; -R: rearranged.

**Table 1 jcm-10-03792-t001:** Clinical implementation of high-throughput sequencing.

Platform	Capability	Cost	Detectable Subtypes	Difficult Subtypes
WTS (RNAseq)	Fusion chimeras Gene expression profiling Mutant allele expression Alternative splicing analysis (BCR/TCR rearrangements) (Sequence mutations) (Copy number analysis)	Moderate	B-ALL ETV6-RUNX1; KMT2A; TCF3-PBX1; BCR-ABL1; DUX4; MEF2D; ZNF384/362 NUTM1; HLF; BCL2/MYC; PAX5alt; ZEB2/CEBPE; -like subtypes	B-ALL Aneuploidies
			T-ALL HOXA (*KMT2A*-R, *PICALM-MLLT10*, *SET-NUP214*); SPI1; NKX2-1; TAL1 (*STIL-TAL1*)	T-ALL BCL11B; TLX1/3; LMO1/2; HOXA (others); TAL1 (others); T-other
WGS	Sequence mutations Structural variants Copy number analysis (BCR/TCR rearrangements) (GWAS)	High	B-ALL Aneuploidies; ETV6-RUNX1; KMT2A; TCF3-PBX1; BCR-ABL1; DUX4; MEF2D; ZNF384/362; NUTM1; HLF; BCL2/MYC; PAX5 P80R; IKZF1 N159Y; ZEB2/CEBPE; Sequence and structural alterations in Ph-like ALL	B-ALL -like subtypes; Part of PAX5alt
			T-ALL BCL11B; TLX1/3; LMO1/2; HOXA; SPI1; NKX2-1; TAL1	T-ALL T-other
WES	Sequence mutations (coding) Structural variants (coding) Copy number analysis	Moderate	B-ALL (Aneuploidies) PAX5 P80R IKZF1 N159Y Sequence mutations in Ph-like ALL (e.g., JAK1/2/3, Ras)	Most of other B-ALL and T-ALL subtypes
Targeted sequencing (DNA and/or RNA)	Fusion chimeras (targeted) Gene expression (targeted) Sequence mutations (targeted) Structural variants (targeted) (Copy number analysis)	Low	Targeted alterations	Non-targeted alterations

The parenthesis in “Capability” indicates analyses in development. Abbreviations: WTS: whole transcriptome sequencing; BCR: B-cell receptor; TCR: T-cell receptor; WGS: whole genome sequencing; GWAS: genome wide association study; WES: whole exome sequencing; -R: rearranged.

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
