# Peer review of "Biologic and Therapeutic Implications of Genomic Alterations in Acute Lymphoblastic Leukemia"

_jcm, 2021, doi:10.3390/jcm10173792_

Round 1

Reviewer 1 Report

The review by Iacobucci et al. presents a comprehensive and timely overview of the different subtypes of B- and T-ALL based on their underlying genetic alterations. There is also a good description of the leukemic phenotypes, and the subtypes are helpfully correlated with prevalence in different age groups and treatment options. Overall, this review will be of great interest to the community, but requires some revisions to help improve the quality of the manuscript.

Specific comments:

  1. My main criticism is that Figure 2 is not particularly helpful or informative. It was just taken straight out of a recent publication of the authors, but does not provide any more information than a simple list of the T-ALL subtypes as shown on the right-hand side of that figure. The inclusion of AML and MPAL types is also unnecessary. It should really be replaced with a figure more akin to Figure 1 that links subtypes with prevalence and prognosis. A diagram outlining T-cell differentiation stages would also be very helpful in this context since the correlation with stage of arrest in T-ALL has been stressed.
  2. The sentence in lines 74/75 on page 2 needs to be slightly rephrased to something like “…with nonrandom gain of typically chromosome X, 4, 74 6, 10, 14, 17, 18, and 21) is…”, as it otherwise suggests that there was a gain of up to 21 chromosomes.
  3. The sentence in lines 77/78 on page 2 needs clarifying: “Alterations involving the Ras pathway (KRAS, NRAS, FTL3, PTPN11) and epigenetic 77 modifiers (CREBBP, WHSC1) are frequent genetic events” – Do these represent new subgroups? Are these affected due to the hyperdiploidy (i.e. more copy numbers of these)? Are there mutations in these genes/pathways that are secondary to the hyperdiploidy? A similar clarification is needed for the RAS mutations mentioned in line 91.
  4. The abbreviation of the term “adolescent and young adult (AYA)” is defined on three different occasions in the manuscript text.
  5. The abbreviation ‘AYA’ needs to be defined in the Figure 1 legends.
  6. The whole manuscript needs another round of careful proof-reading as there were many typographical, grammar and formatting mistakes. For example, there is an unfinished sentence in line 244 on page 5.

Author Response

Review 1

Comments and Suggestions for Authors

The review by Iacobucci et al. presents a comprehensive and timely overview of the different subtypes of B- and T-ALL based on their underlying genetic alterations. There is also a good description of the leukemic phenotypes, and the subtypes are helpfully correlated with prevalence in different age groups and treatment options. Overall, this review will be of great interest to the community, but requires some revisions to help improve the quality of the manuscript.

Specific comments:

  1. My main criticism is that Figure 2 is not particularly helpful or informative. It was just taken straight out of a recent publication of the authors, but does not provide any more information than a simple list of the T-ALL subtypes as shown on the right-hand side of that figure. The inclusion of AML and MPAL types is also unnecessary. It should really be replaced with a figure more akin to Figure 1 that links subtypes with prevalence and prognosis. A diagram outlining T-cell differentiation stages would also be very helpful in this context since the correlation with stage of arrest in T-ALL has been stressed.

Response: We have provided a new figure 2 that is more informative.

  1. The sentence in lines 74/75 on page 2 needs to be slightly rephrased to something like “…with nonrandom gain of typically chromosome X, 4, 74 6, 10, 14, 17, 18, and 21) is…”, as it otherwise suggests that there was a gain of up to 21 chromosomes.

Response: This has been rewritten

  1. The sentence in lines 77/78 on page 2 needs clarifying: “Alterations involving the Ras pathway (KRAS, NRAS, FTL3, PTPN11) and epigenetic 77 modifiers (CREBBP, WHSC1) are frequent genetic events” – Do these represent new subgroups? Are these affected due to the hyperdiploidy (i.e. more copy numbers of these)? Are there mutations in these genes/pathways that are secondary to the hyperdiploidy? A similar clarification is needed for the RAS mutations mentioned in line 91.

Response: This has been rewritten to reflect these are common concomitant genetic events in hyperdiploid ALL (rather than new subgroups). We have also added a comment in the subsequent paragraph on hypodiploid ALL to reflect the similar gene expression profile and patterns of co-mutation between high hyperdiploid and near haploid ALL.

  1. The abbreviation of the term “adolescent and young adult (AYA)” is defined on three different occasions in the manuscript text.

Response: fixed

  1. The abbreviation ‘AYA’ needs to be defined in the Figure 1 legends.

Response: added

  1. The whole manuscript needs another round of careful proof-reading as there were many typographical, grammar and formatting mistakes. For example, there is an unfinished sentence in line 244 on page 5.

Response: We have proofed the revision.

Reviewer 2 Report

This review aims to describe the current genomic landscape of B- and T-ALL, highlighting their genetic characterization and diagnostic classification, clinical features, and therapeutic implications. Moreover, this review emphasizes the utilization of next generation sequencing to identify novel risk-associated ALL subtypes, including the classical subtypes characterized by germline and somatic genetic alterations that converge on distinct gene expression profiles. Nevertheless, and in spite of the significant amount of work performed, some important issues have to be consider.

  1. In my opinion, the main novelty of this review should be the analysis of New Generation Sequencing approaches to identify the current genetic characterization and diagnostic classification in ALL. Nevertheless, the review is mainly focused in describing the classical mutations identified by FISH and karyotype. In the same way, no clinical features are included in this work.
  2. On the other hand, I have done a quick search in Pubmed, and I have found a review identical to this work. So, could the authors explain the novelty of their review compared to others published recently, in order to be consider for publication?
  • Nicoletta Coccaro et al. “Next-Generation Sequencing in Acute Lymphoblastic Leukemia”. Int J Mol Sci. 2019 Jun 15;20(12):2929. doi: 10.3390/ijms20122929.
  1. The literature used in this review includes references from the last 10 years, and only around a 30% are references from the last 5 years. In my opinion and based on the purpose of this review focusing on the compilation of NGS approaches to identified novel risk-associated ALL subtypes, could the authors explain why they use such outdated references?
  2. In my opinion, the review contains a lot of information and more tables or figures should be included to make it easier for readers to read and understand.

Author Response

This review aims to describe the current genomic landscape of B- and T-ALL, highlighting their genetic characterization and diagnostic classification, clinical features, and therapeutic implications. Moreover, this review emphasizes the utilization of next generation sequencing to identify novel risk-associated ALL subtypes, including the classical subtypes characterized by germline and somatic genetic alterations that converge on distinct gene expression profiles. Nevertheless, and in spite of the significant amount of work performed, some important issues have to be consider.

  1. In my opinion, the main novelty of this review should be the analysis of New Generation Sequencing approaches to identify the current genetic characterization and diagnostic classification in ALL. Nevertheless, the review is mainly focused in describing the classical mutations identified by FISH and karyotype. In the same way, no clinical features are included in this work.

Response: This is a solicited review on the genetics and biology of ALL. We have included comments on clinical features, outcome and therapeutic intervention throughout, and have added a section on diagnostics.

  1. On the other hand, I have done a quick search in Pubmed, and I have found a review identical to this work. So, could the authors explain the novelty of their review compared to others published recently, in order to be consider for publication?

  • Nicoletta Coccaro et al. “Next-Generation Sequencing in Acute Lymphoblastic Leukemia”. Int J Mol Sci. 2019 Jun 15;20(12):2929. doi: 10.3390/ijms20122929.

Response: It does not appear that the authors of that manuscript have had a substantive role in leading or performing key studies in both reviews, to which they refer. We have provided a review on the topic requested by the editor.

  1. The literature used in this review includes references from the last 10 years, and only around a 30% are references from the last 5 years. In my opinion and based on the purpose of this review focusing on the compilation of NGS approaches to identified novel risk-associated ALL subtypes, could the authors explain why they use such outdated references?

Response: see above; and where pertinent we have included recent references.

  1. In my opinion, the review contains a lot of information and more tables or figures should be included to make it easier for readers to read and understand

Response: we have added additional figures and tables.

Round 2

Reviewer 1 Report

The authors have addressed all of my concerns and revised the manuscript to a high standard. The new Figure 2 is excellent and the addition of section 4 and Table 1 very useful. I only have one minor request: to add the definition of the abbreviation 'WES' to the legends of Table 1.

Author Response

The definition of WES has been added to the notes for Table 1.

Reviewer 2 Report

Comments to the Author:

I commend the effort made by the authors to revise this manuscript. The authors have answered correctly all my questions and they have added more figures and tables into the review, as I suggested to them. The authors have included more up-to-date references where appropriate. In the same way, they have included a new section, “4. Implications for diagnosis” and Table 1, that I think it are very informative and sums up quite well the current landscape with regard to the diagnosis of the ALL. I consider that now the review is more solid and it has improved considerably for its publication. However, I still have some issues that should be addressed.

General comments:

  1. In Introduction section, the authors say that this review describes “the current genomic landscape of B- and T-ALL, highlighting their genetic characterization and diagnostic classification, clinical features, and therapeutic implications”. In my first review, I suggested the authors to include more information concerning with clinical features, because the review was poor in this data. I still think that the review should include more clinical features other than blood cell counts or immunophenotypic features. Similarly, there are some sections that lack this information, such as 2.1.1 Subtypes with chromosomal aneuploidy.
  2. The authors should check the spelling as there are many grammatical errors throughout the review.

Author Response

The review already contains extensive additional clinical data apart from blood counts and immunophenotype, such as MRD, outcome and therapeutic approaches/opportunities.

Here, in this second revision, additional clinical data have been added to 2.1.1 “Subtypes with chromosomal aneuploidy”. 

We have made additional corrections and further proofed the document